# CONTROLVIDEO: TRAINING-FREE CONTROLLABLE TEXT-TO-VIDEO GENERATION

**Yabo Zhang**[1] **Yuxiang Wei**[1] **Dongsheng Jiang**[2] **Xiaopeng Zhang**[2] **Wangmeng Zuo**[1] (✉) **Qi Tian**[2]

[1]Harbin Institute of Technology [2]Huawei Cloud

## ABSTRACT

Text-driven diffusion models have unlocked unprecedented abilities in image generation, whereas their video counterpart lags behind due to the excessive training cost. To avert the training burden, we propose a training-free ControlVideo to produce high-quality videos based on the provided text prompts and motion sequences. Specifically, ControlVideo adapts a pre-trained text-to-image model (*i.e.*, ControlNet) for controllable text-to-video generation. To generate continuous videos without flicker effects, we propose an *interleaved-frame smoother* to smooth the intermediate frames. In particular, interleaved-frame smoother splits the whole video with successive three-frame clips, and stabilizes each clip by updating the middle frame with the interpolation among other two frames in latent space. Furthermore, a *fully cross-frame interaction* mechanism is exploited to further enhance the frame consistency, while a *hierarchical sampler* is employed to produce long videos efficiently. Extensive experiments demonstrate that our ControlVideo outperforms the state-of-the-arts both quantitatively and qualitatively. It is worth noting that, thanks to the efficient designs, ControlVideo could generate both short and long videos within several minutes using one NVIDIA 2080Ti. Code and videos are available at this link.

## 1 INTRODUCTION

Large-scale diffusion models have made a tremendous breakthrough on text-to-image synthesis (Nichol et al., 2021; Rombach et al., 2022; Balaji et al., 2022; Ramesh et al., 2022; Saharia et al., 2022) and their creative applications (Gal et al., 2022; Wei et al., 2023; Ni et al., 2022; Hertz et al., 2022). Several studies (Ho et al., 2022b;a; Singer et al., 2022; Esser et al., 2023; Hong et al., 2022) attempt to replicate this success in the video counterpart, *i.e.*, modeling higher-dimensional complex video distributions in the wild world. However, training such a text-to-video model requires massive amounts of high-quality videos and computational resources, which limits further research and applications by relevant communities.

In this work, we study a new and efficient form to avert the excessive training requirements: *controllable text-to-video generation with text-to-image models*. As shown in Fig. 1, our method, termed ControlVideo, takes textual description and motion sequence (*e.g.*, depth or edge maps) as conditions to generate videos. Instead of learning the video distribution from scratch, ControlVideo adapts the pre-trained text-to-image models (*e.g.*, ControlNet (Zhang & Agrawala, 2023)) for high-quality video generation. With the structural information from motion sequence and the superior generation capability of image models, it is feasible to produce a vivid video without additional training.

However, as shown in Fig. 1, due to the lack of temporal interaction, individually producing each frame with ControlNet (Zhang & Agrawala, 2023) fails to ensure both (i) *frame consistency* and (ii) *video continuity*. Frame consistency requires all frames to be generated with a coherent appearance, while video continuity ensures smooth transitions between frames. Tune-A-Video (Wu et al., 2022b) and Text2Video-Zero (Khachatryan et al., 2023) facilitate appearance consistency by extending self-attention to sparser cross-frame attention. Nonetheless, such a cross-frame interaction is not sufficient to guarantee video continuity, and visible flickers appear in their synthesized videos (as shown in Fig. 1 and corresponding videos).

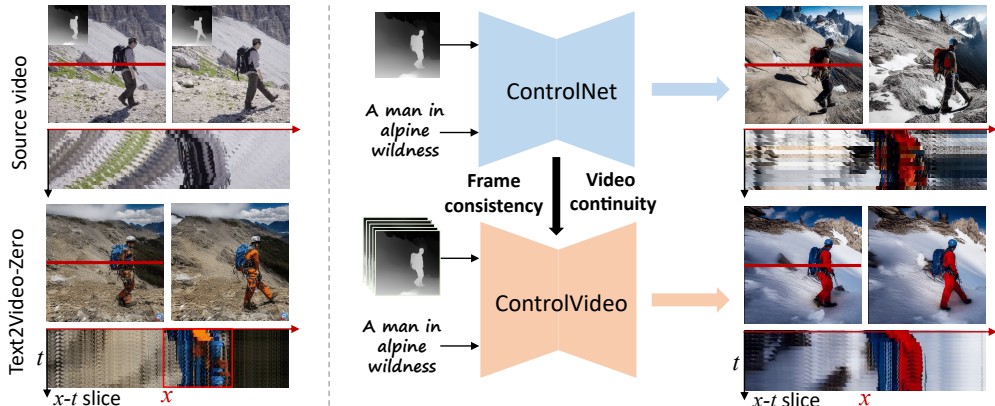

Figure 1: **Training-free controllable text-to-video generation. Left:** We visualize the frames and $x$-$t$ slice (pixels in red line of original frame) of Text2Video-Zero, and observe visible discontinuity in $x$-$t$ slice. **Right:** ControlVideo, adapted from ControlNet, achieves more continuous $x$-$t$ slice across time, along with improved appearance consistency than Text2Video-Zero. *See videos for better view.*

Intuitively, a continuous video could be considered as multiple continuous three-frame clips, so the problem of *ensuring the video continuity is converted to ensuring all three-frame clips continuous*. Driven by this analysis, we propose an *interleaved-frame smoother* to enable continuous video generation. Specifically, interleaved-frame smoother divides all three-frame clips into even and odd clips based on indices of middle frames, and separately smooths out their corresponding latents at different denoising steps. To stabilize the latent of each clip, we first convert it to predicted RGB frames with DDIM, followed by replacing the middle frame with the interpolated frame. Note that, the smoother is only applied at a few timesteps, and the quality and individuality of interpolated frames can be well retained by the following denoising steps.

We further investigate the cross-frame mechanisms in terms of effectiveness and efficiency. Firstly, we explore *fully cross-frame interaction* that concatenates all frames to become a "larger image", and first empirically demonstrate its superior consistency and quality than sparser counterparts (see Sec. 4.4). Secondly, applying existing cross-frame mechanisms for long-video generation suffers from either heavy computational burden or long-term inconsistency. Therefore, a *hierarchical sampler* is presented to produce a long video in a top-down way. In specific, it pre-generates the key frames with fully cross-frame attention for long-range coherence, followed by efficiently generating the short clips conditioned on pairs of key frames.

We conduct the experiments on extensively collected motion-prompt pairs, and show that ControlVideo outperforms alternative competitors qualitatively and quantitatively. Thanks to the efficient designs, ControlVideo produces short and long videos in several minutes using one NVIDIA 2080Ti.

In summary, our contributions are presented as follows:

- We propose training-free ControlVideo with interleaved-frame smoother for consistent and continuous controllable text-to-video generation.
- Interleaved-frame smoother alternately smooths out the latents of three-frame clips, effectively stabilizing the entire video during sampling.
- We empirically demonstrate the superior consistency and quality of fully cross-frame interaction, while presenting a hierarchical sampler for long-video generation in commodity GPUs.

## 2 BACKGROUND

**Latent diffusion model** (LDM) (Rombach et al., 2022) is an efficient variant of diffusion models (Ho et al., 2020) by applying the diffusion process in the latent space. LDM uses an encoder to compress an image $x$ into latent code $z = (x)$. It learns the distribution of image latent codes $z_0 \sim p_{data}(z_0)$ in a DDPM formulation (Ho et al., 2020), including a forward and a backward process. The forward diffusion process gradually adds gaussian noise at each timestep $t$ to obtain $z_t$:

$$q(z_t|z_{t-1}) = \mathcal{N}(z_t; \sqrt{1-\beta_t}z_{t-1}, \beta_t I), \tag{1}$$

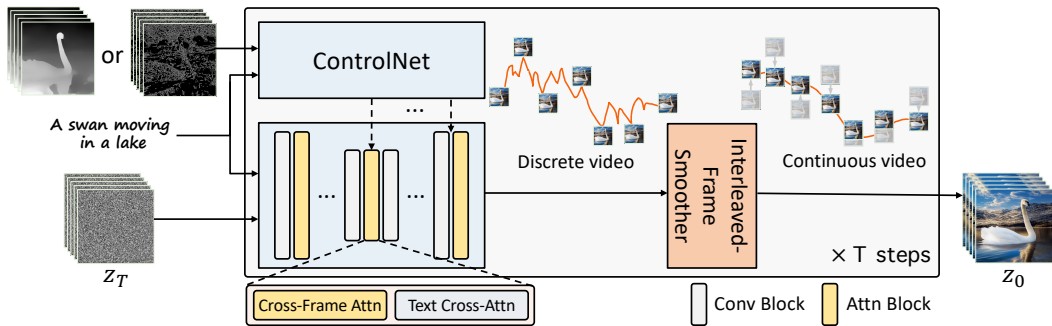

Figure 2: **Overview of ControlVideo.** For consistency in appearance, ControlVideo adapts ControlNet to the video counterpart by adding cross-frame interaction into self-attention modules. To further improve video continuity, interleaved-frame smoother is introduced to stabilize video latents during denosing (see Alg. 1 for details).

where $\{\beta_t\}_{t=1}^{T}$ are the scale of noises, and $T$ denotes the number of diffusion timesteps. The backward denoising process reverses the above diffusion process to predict less noisy $z_{t-1}$:

$$p_\theta(z_{t-1}|z_t) = \mathcal{N}(z_{t-1}; \mu_\theta(z_t, t), \Sigma_\theta(z_t, t)). \qquad (2)$$

The $\mu_\theta$ and $\Sigma_\theta$ are implemented with a denoising model $\epsilon_\theta$ with learnable parameters $\theta$. When generating new samples, we start from $z_T \sim \mathcal{N}(0, 1)$ and employ DDIM sampling to predict $z_{t-1}$ of previous timestep:

$$z_{t-1} = \sqrt{\alpha_{t-1}} \underbrace{\left( \frac{z_t - \sqrt{1-\alpha_t}\epsilon_\theta(z_t, t)}{\sqrt{\alpha_t}} \right)}_{\text{`` predicted } z_0\text{''}} + \underbrace{\sqrt{1-\alpha_{t-1}} \cdot \epsilon_\theta(z_t, t)}_{\text{``direction pointing to } z_t\text{''}}, \qquad (3)$$

where $\alpha_t = \prod_{i=1}^{t}(1 - \beta_i)$. We use $z_{t\to0}$ to represent "predicted $z_0$" at timestep $t$ for simplicity. Note that we use Stable Diffusion (SD) $\epsilon_\theta(z_t, t, \tau)$ as our base model, which is an instantiation of text-guided LDMs pre-trained on billions of image-text pairs. $\tau$ denotes the text prompt.

**ControlNet** (Zhang & Agrawala, 2023) enables SD to support more controllable input conditions during text-to-image synthesis, *e.g.*, depth maps, poses, edges, *etc*. The ControlNet uses the same U-Net (Ronneberger et al., 2015) architecture as SD and finetunes its weights to support task-specific conditions, converting $\epsilon_\theta(z_t, t, \tau)$ to $\epsilon_\theta(z_t, t, c, \tau)$, where $c$ denotes additional conditions. To distinguish the U-Net architectures of SD and ControlNet, we denote the former as the *main U-Net* while the latter as the *auxiliary U-Net*.

## 3 CONTROLVIDEO

Controllable text-to-video generation aims to produce a video of length $N$ conditioned on motion sequences $c = \{c^i\}_{i=0}^{N-1}$ and a text prompt $\tau$. As illustrated in Fig. 2, we propose ControlVideo with *interleaved-frame smoother* towards consistent and continuous video generation. ControlVideo, adapted from ControlNet, adds cross-frame interaction to self-attention modules for frame consistency (in Sec. 3.1). To ensure video continuity, *interleaved-frame smoother* divides all three-frame clips into even and odd clips, and separately smooths out their corresponding latents at different denoising steps (in Sec. 3.2). Finally, we further investigate the cross-frame mechanisms in terms of effectiveness and efficiency, including fully cross-frame interaction and hierarchical sampler (in Sec. 3.3).

### 3.1 PRELIMINARY

The main challenge of adapting text-to-image models to the video counterpart is to ensure temporal consistency. Leveraging the controllability of ControlNet, motion sequences could provide coarse-level consistency in structure. Nonetheless, due to the lack of temporal interaction, individually producing each frame with ControlNet leads to drastic inconsistency in appearance (see row 2 in

---

**Algorithm 1** Interleaved-frame smoother

---

**Require:** $z_t = \{z_t^i\}_{i=0}^{N-1}$, $c = \{c^i\}_{i=0}^{N-1}$, $\tau$, timestep $t$.

1: $z_{t\to0} \leftarrow \frac{z_t - \sqrt{1-\alpha_t}\epsilon_\theta(z_t,t,c,\tau)}{\sqrt{\alpha_t}}$.        ▷ *predict clean latents*

2: $x_{t\to0} \leftarrow \mathcal{D}(z_{t\to0})$; $\tilde{x}_{t\to0} \leftarrow x_{t\to0}$        ▷ *convert latents to RGB space*

3: **if** $(t \mod 2) = 0$ **then**      ▷ *smooth all even three-frame clips $(\tilde{x}_{t\to0}^{2k-1}, \tilde{x}_{t\to0}^{2k}, \tilde{x}_{t\to0}^{2k+1})$*

4:      **for** $k$ from $0$ to $N/2$ **do**

5:         $\tilde{x}_{t\to0}^{2k} \leftarrow \text{Interpolate}(x_{t\to0}^{2k-1}, x_{t\to0}^{2k+1})$

6: **else if** $(t \mod 2) = 1$ **then**      ▷ *smooth all odd three-frame clips $(\tilde{x}_{t\to0}^{2k}, \tilde{x}_{t\to0}^{2k+1}, \tilde{x}_{t\to0}^{2k+2})$*

7:      **for** $k$ from $0$ to $N/2$ **do**

8:         $\tilde{x}_{t\to0}^{2k+1} \leftarrow \text{Interpolate}(x_{t\to0}^{2k}, x_{t\to0}^{2k+2})$

9: $\tilde{z}_{t\to0} \leftarrow \mathcal{E}(\tilde{x}_{t\to0})$        ▷ *convert frames to latent space*

10: $z_{t-1} \leftarrow \sqrt{\alpha_{t-1}}\tilde{z}_{t\to0} + \sqrt{1-\alpha_{t-1}} \cdot \epsilon_\theta(z_t,t,c,\tau)$.        ▷ *predict less noisy latent*

11: **return** $z_{t-1}$

---

Fig. 5). Similar to previous works (Wu et al., 2022b; Khachatryan et al., 2023), we also extend original self-attention of SD U-Net to cross-frame attention, so that the video content could be temporally shared via inter-frame interaction.

In specific, ControlVideo inflates the main U-Net from Stable Diffusion along the temporal axis, while keeping the auxiliary U-Net from ControlNet. Analogous to (Ho et al., 2022b; Wu et al., 2022b; Khachatryan et al., 2023), it directly converts 2D convolution layers to 3D counterpart by replacing $3 \times 3$ kernels with $1 \times 3 \times 3$ kernels. Self-attention is converted to cross-frame attention by querying from other frames as:

$$\text{Attention}(Q, K, V) = \text{Softmax}(\frac{QK^T}{\sqrt{d}}) \cdot V, \text{ where } Q = W^Q z_t^i, \ K = W^K \tilde{z}_t, \ V = W^V \tilde{z}_t, \quad (4)$$

where $W^Q$, $W^K$, and $W^V$ project $z_t$ into query, key, and value, respectively. $z_t^i$ and $\tilde{z}_t$ denote $i$th latent frame and the latents of reference frames at timestep $t$. We will discuss the choices of cross-frame mechanisms (*i.e.*, reference frames) in Sec. 3.3

## 3.2 INTERLEAVED-FRAME SMOOTHER

Albeit cross-frame interaction promisingly keeps frame consistency in appearance, they are still visibly flickering in structure. Discrete motion sequences only ensure coarse-level structural consistency, not sufficient to keep the continuous inter-frame transition. Intuitively, a continuous video could be considered as multiple continuous three-frame clips, so we simplify the problem of ensuring the video continuity to ensuring all three-frame clips continuous.

Inspired by this, we propose an *interleaved-frame smoother* to enable continuous video generation. In Alg. 1, interleaved-frame smoother divides all three-frame clips into even and odd clips based on indices of middle frames, and individually smooths their corresponding latents at different timesteps. To stabilize the latent of each clip, we first convert it to predicted RGB frames with DDIM, following by replacing middle frame with the interpolated frame.

Specifically, at timestep $t$, we first predict the clean video latent $z_{t\to0}$ according to $z_t$:

$$z_{t\to0} = \frac{z_t - \sqrt{1-\alpha_t}\epsilon_\theta(z_t,t,c,\tau)}{\sqrt{\alpha_t}}. \quad (5)$$

After projecting $z_{t\to0}$ into a RGB video $x_{t\to0} = \mathcal{D}(z_{t\to0})$, we convert it to a more smoothed video $\tilde{x}_{t\to0}$ by replacing each middle frame with the interpolated one. Based on smoothed video latent $\tilde{z}_{t\to0} = \mathcal{E}(\tilde{x}_{t\to0})$, we compute the less noisy latent $z_{t-1}$ following DDIM denoising in Eq. 3:

$$z_{t-1} = \sqrt{\alpha_{t-1}}\tilde{z}_{t\to0} + \sqrt{1-\alpha_{t-1}} \cdot \epsilon_\theta(z_t,t,c,\tau). \quad (6)$$

We note that the above process is only performed at a few intermediate timesteps, the individuality and quality of interpolated frames are also well retained by the following denoising steps. Additionally, the newly computational burden can be negligible (See Table 3).

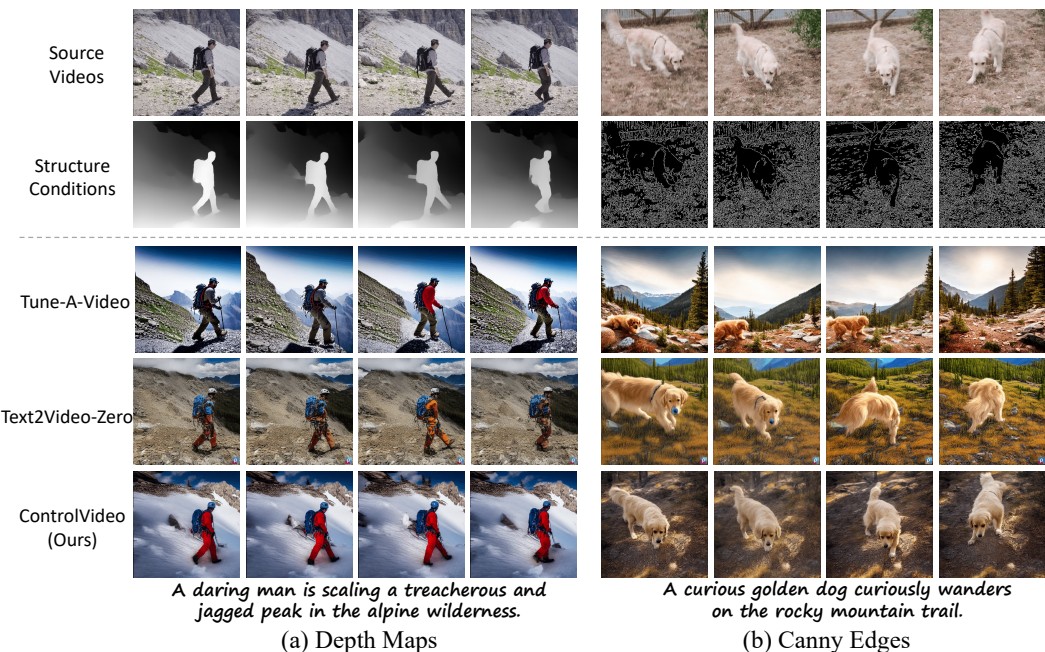

A daring man is scaling a treacherous and jagged peak in the alpine wilderness.

(a) Depth Maps

A curious golden dog curiously wanders on the rocky mountain trail.

(b) Canny Edges

Figure 3: **Qualitative comparisons conditioned on depth maps and canny edges.** Our ControlVideo produces videos with better (a) appearance consistency and (b) video quality than others. In contrast, Tune-A-Video fails to inherit structures from source videos, while Text2Video-Zero brings visible artifacts in large motion videos. *See videos at qualitative comparisons.*

### 3.3 CROSS-FRAME MECHANISMS FOR EFFECTIVENESS AND EFFICIENCY

**Fully cross-frame interaction.** Previous works (Wu et al., 2022b; Khachatryan et al., 2023) usually replace self-attention with sparser cross-frame mechanisms, *e.g.*, taking the reference frames as first or previous frames. Such mechanisms will increase the discrepancy between the query and key in self-attention modules, resulting in the degradation of video quality and consistency. In contrast, fully cross-frame interaction considers all frames as reference (*i.e.,* becoming a "large image"), so has a less generation gap with text-to-image models. We conduct comparison experiments on above mechanisms in Fig. 5 and Table 3. Despite slightly more computational burden, fully cross-frame interaction empirically shows better consistency and quality than the sparser counterparts.

**Hierarchical sampler.** Applying existing cross-frame mechanisms for long-video generation suffers from either heavy computational burden or long-term inconsistency, limiting the practicability of ControlVideo. For more efficient long-video synthesis, we introduce a hierarchical sampler to produce a long video clip-by-clip, which is implemented with two types of cross-frame mechanisms. At each timestep, a long video $z_t = \{z_t^i\}_{i=0}^{N-1}$ is separated into multiple short video clips with the selected key frames $z_t^{key} = \{z_t^{kN_c}\}_{k=0}^{\frac{N}{N_c}}$, where each clip is of length $N_c - 1$ and the $k$th clip is denoted as $\widehat{z}_t^k = \{z_t^j\}_{j=kN_c+1}^{(k+1)N_c-1}$. Then, we pre-generate the key frames with *fully* cross-frame attention for long-range coherence, where reference frames are $z_t^{key} = \{z_t^{kN_c}\}_{k=0}^{\frac{N}{N_c}}$. Conditioned on each pair of key frames, *i.e.*, reference frames as $\{z_t^{kN_c}, z_t^{(k+1)N_c}\}$, we sequentially synthesize their corresponding clip $\widehat{z}_t^k$ holding the holistic consistency.

## 4 EXPERIMENTS

### 4.1 EXPERIMENTAL SETTINGS

**Implementation details.** ControlVideo is adapted from ControlNet [1] (Zhang & Agrawala, 2023), and our interleaved-frame smoother employs a lightweight RIFE (Huang et al., 2022) to interpolate

---

[1]https://huggingface.co/lllyasviel/ControlNet

Table 1: **Quantitative comparisons** of ControlVideo with other methods. We evaluate them on 125 motion-prompt pairs in terms of consistency, and the best results are **bolded**.

| METHOD | Structure Condition | FC ($\times 10^{-2}$) | PC ($\times 10^{-2}$) | WE ($\times 10^{-2}$) |
|---|---|---|---|---|
| Tune-A-Video Wu et al. (2022b) | DDIM Inversion | 94.53 | 31.57 | 18.16 |
| Text2Video-Zero Khachatryan et al. (2023) | Canny Edge | 95.17 | 30.74 | 8.76 |
| **ControlVideo (ours)** | Canny Edge | **96.83** | **30.75** | **2.75** |
| Text2Video-Zero Khachatryan et al. (2023) | Depth Map | 95.99 | 31.69 | 10.36 |
| **ControlVideo (ours)** | Depth Map | **97.22** | **31.81** | **5.81** |

the middle frame of each three-frame clip. The synthesized short videos are of length 15, while the long videos usually contain about 100 frames. Unless otherwise noted, their resolution is both $512 \times 512$. During sampling, we adopt DDIM sampling (Song et al., 2020a) with 50 timesteps, and interleaved-frame smoother is performed on predicted RGB frames at timesteps $\{30, 31\}$ by default. With the efficient implementation of xFormers (Lefaudeux et al., 2022), ControVideo could produce both short and long videos with one NVIDIA RTX 2080Ti in about 2 and 10 minutes, respectively.

**Datasets.** To evaluate our ControlVideo, we collect 25 object-centric videos from DAVIS dataset (Pont-Tuset et al., 2017) and manually annotate their source descriptions. Then, for each source description, ChatGPT (OpenAI, 2022) is utilized to generate five editing prompts automatically, resulting in 125 video-prompt pairs in total. Finally, we employ Canny and MiDaS DPT-Hybrid model (Ranftl et al., 2020) to estimate the edges and depth maps of source videos, and form 125 motion-prompt pairs as our evaluation dataset. More details are provided in Appendix A.

**Metrics.** We evaluate the video quality from three perspectives. (i) Frame consistency (FC): the average cosine similarity between all pairs of consecutive frames, *and* (ii) Prompt consistency (PC): the average cosine similarity between input prompt and all video frames. (iii) Warping error (WE) (Lai et al., 2018): the average error between all frames and their warped frames using optical flow.

**Baselines.** We compare our ControlVideo with three publicly available methods: (i) Tune-A-Video (Wu et al., 2022b) extends Stable Diffusion to the video counterpart by finetuning it on a source video. During inference, it uses the DDIM inversion codes of source videos to provide structure guidance. (ii) Text2Video-Zero (Khachatryan et al., 2023) is based on ControlNet, and employs the first-only cross-frame attention on Stable Diffusion without finetuning. (iii) Follow-Your-Pose (Ma et al., 2023) is initialized with Stable Diffusion, and is finetuned on LAION-Pose (Ma et al., 2023) to support human pose conditions. After that, it is trained on millions of videos (Xue et al., 2022) to enable temporally-consistent video generation.

## 4.2 QUALITATIVE AND QUANTITATIVE COMPARISONS

**Qualitative results.** Fig. 3 first illustrates the visual comparisons of synthesized videos conditioned on both (a) depth maps and (b) canny edges. As shown in Fig. 3 (a), our ControlVideo demonstrates better consistency in both appearance and structure than alternative competitors. Tune-A-Video fails to keep the temporal consistency of both appearance and fine-grained structure, *e.g.*, `the color of coat and the structure of road`. With the motion information from depth maps, Text2Video-Zero achieves promising consistency in structure, but still struggles with incoherent appearance in videos *e.g.*, `the color of coat`. Besides, ControlVideo also performs more robustly when dealing with large motion inputs. As illustrated in Fig. 3 (b), Tune-A-Video ignores the structure information from source videos. Text2Video-Zero adopts the first-only cross-frame mechanism to trade off frame quality and appearance consistency, and generates later frames with visible artifacts. In contrast, with the proposed fully cross-frame mechanism and interleaved-frame smoother, our ControlVideo can handle large motion to generate high-quality and consistent videos.

Fig. 4 further shows the comparison conditioned on human poses. From Fig. 4, Tune-A-Video only maintains the coarse structures of the source video, *i.e.*, `human position`. Text2Video-Zero and Follow-Your-Pose produce video frames with inconsistent appearance, *e.g.*, `changing faces of iron man` (in row 4) or `disappearing objects in the background` (in row 5). In comparison, our ControlVideo performs more consistent video generation, demonstrating its superiority. More qualitative comparisons are provided in Appendix D.

Table 2: **User preference study.** The numbers denote the percentage of raters who favor the videos synthesized by our ControlVideo over other methods.

| Method Comparison | Video Quality | Temporal Consistency | Text Alignment |
|---|---|---|---|
| Ours vs. Tune-A-Video Wu et al. (2022b) | 73.6% | 83.2% | 68.0% |
| Ours vs. Text2Video-Zero Khachatryan et al. (2023) | 76.0% | 81.6% | 65.6% |

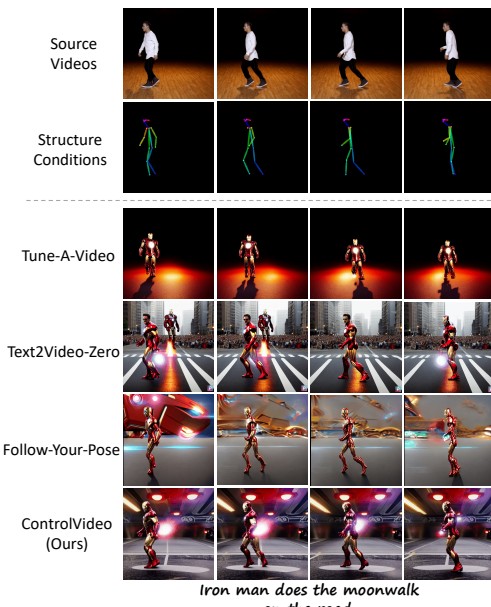

Figure 4: **Qualitative comparisons on poses**. Tune-A-Video only preserves original human positions, while Text2Video-Zero and Follow-Your-Pose produce frames with appearance incoherence. Our ControlVideo achieves better consistency in both structure and appearance. *See videos at qualitative comparisons.*

Figure 5: **Qualitative ablation studies** on cross-frame mechanisms and interleaved-frame smoother. Fully cross-frame interaction produces video frames with higher quality and consistency than other mechanisms, and adding the smoother further enhances the video smoothness. *See corresponding videos for better comparison.*

**Quantitative results.** We have also compared our ControlVideo with existing methods quantitatively on 125 video-prompt pairs. From Table 1, our ControlVideo conditioned on depth outperforms the state-of-the-art methods in terms of all metrics, which is consistent with the qualitative results. In contrast, despite finetuning on a source video, Tune-A-Video still struggles to produce temporally coherent videos. Although conditioned on the same structure information, Text2Video-Zero obtains worse frame consistency and warping error than ControlVideo. For each method, the depth-conditioned models generate videos with higher frame and prompt consistency than the canny-condition counterpart, since depth maps provide smoother motion information.

## 4.3 USER STUDY

We then perform the user study to compare our ControlVideo conditioned on depth maps with other competing methods. In specific, we provide each rater a structure sequence, a text prompt, and synthesized videos from two different methods (in random order). Then we ask them to select the better synthesized videos for each of three measurements: (i) video quality, (ii) temporal consistency throughout all frames, and (iii) text alignment between prompts and synthesized videos. The evaluation set consists of 125 representative structure-prompt pairs. Each pair is evaluated by 5 raters, and we take a majority vote for the final result. From Table 2, the raters strongly favor our synthesized videos from all three perspectives, especially in temporal consistency. On the other hand, Tune-A-Video fails to generate consistent and high-quality videos with only DDIM inversion for structural guidance, and Text2Video-Zero also produces videos with lower quality and coherency.

Table 3: **Quantitative ablation studies** on cross-frame mechanisms and interleaved-frame smoother. The results indicate that our fully cross-frame mechanism achieves better frame consistency than other mechanisms, and the interleaved-frame smoother significantly improves the frame consistency.

| Cross-Frame Mechanism | FC ($\times 10^{-2}$) | PC ($\times 10^{-2}$) | WE ($\times 10^{-2}$) | Time Cost (min) |
|---|---|---|---|---|
| Individual | 89.94 | 30.79 | 20.13 | 1.2 |
| First-only | 94.92 | 30.54 | 8.91 | 1.2 |
| Sparse-Causal | 95.06 | 30.59 | 7.05 | 1.5 |
| Fully | 95.36 | 30.76 | 5.93 | 3.0 |
| Fully + Smoother | **96.83** | **30.79** | **2.75** | 3.5 |

a steamship on the ocean, at sunset, sketch style

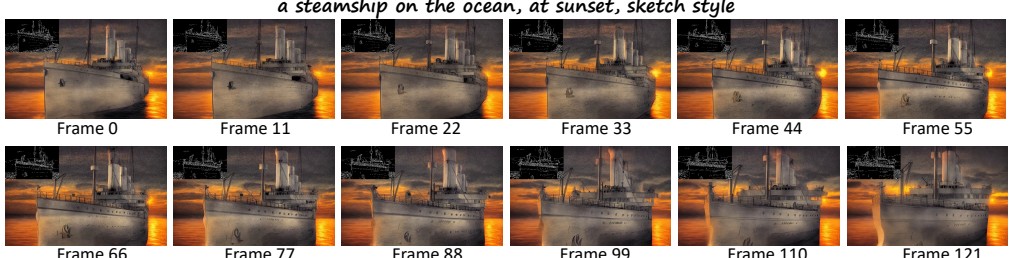

Figure 6: **A long video produced with our hierarchical sampling.** Motion sequences are shown on the top left. Using the efficient sampler, our ControlVideo generates a high-quality long video with the holistic consistency. *See videos at long video generation.*

## 4.4 ABLATION STUDY

**Effect of fully cross-frame interaction.** To demonstrate the effectiveness of the fully cross-frame interaction, we conduct a comparison with the following variants: i) individual: no interaction between all frames, ii) first-only: all frames attend to the first one, iii) sparse-causal: each frame attends to the first and former frames, iv) fully: our fully cross-frame, refer to Sec. 3. Note that, all the above models are extended from ControlNet without any finetuning. The qualitative and quantitative results are shown in Fig. 5 and Table 3, respectively. From Fig. 5, the individual cross-frame mechanism suffers from severe temporal inconsistency, *e.g.*, `colorful and black-and-white frames`. The first-only and sparse-causal mechanisms reduce some appearance inconsistency by adding cross-frame interaction. However, they still produce videos with structural inconsistency and visible artifacts, *e.g.*, `the orientation of the elephant and duplicate nose` (row 3 in Fig. 5). In contrast, due to less generation gap with ControlNet, our fully cross-frame interaction performs better appearance coherency and video quality. Though the introduced interaction brings an extra $1 \sim 2\times$ time cost, it is acceptable for a high-quality video generation.

**Effect of interleaved-frame smoother.** We further analyze the effect of the proposed interleaved-frame smoother. From Table 3 and last two rows of Fig. 5, our interleaved-frame smoother greatly improves the video smoothness, *e.g.*, mitigating structural flickers in red boxes. We provide more ablation studies on the timestep choices of the smoother in Appendix C and ablation studies.

## 4.5 EXTENSION TO LONG-VIDEO GENERATION

Producing a long video usually requires an advanced GPU with high memory. With the proposed hierarchical sampler, our ControlVideo achieves long video generation (more than 100 frames) in a memory-efficient manner. As shown in Fig. 6, our ControlVideo can produce a long video with consistently high quality. Notably, benefiting from our efficient sampling, it only takes approximately ten minutes to generate 100 frames with resolution $512 \times 512$ in one NVIDIA RTX 2080Ti. More visualizations of long videos can be found in Appendix D.

## 5 RELATED WORK

**Text-to-image synthesis.** Through pre-training on billions of image-text pairs, large-scale generative models (Nichol et al., 2021; Balaji et al., 2022; Saharia et al., 2022; Ramesh et al., 2022; Rombach et al., 2022; Ramesh et al., 2021; Chang et al., 2023; Ding et al., 2021; 2022; Yu et al., 2022; Sauer et al., 2023; Kang et al., 2023; Huang et al., 2023) have made remarkable progress in creative and photo-realistic visual generation. Various frameworks have been explored to enhance image quality, including GANs (Goodfellow et al., 2020; Sauer et al., 2023; Kang et al., 2023), autoregressive models (Nichol et al., 2021; Chang et al., 2023; Ding et al., 2021; 2022; Yu et al., 2022), and diffusion models (Ho et al., 2020; Balaji et al., 2022; Saharia et al., 2022; Ramesh et al., 2022; Rombach et al., 2022). Among these generative models, diffusion-based models are well open-sourced and popularly applied to several downstream tasks, such as image editing (Hertz et al., 2022; Meng et al., 2021) and customized generation (Gal et al., 2022; Wei et al., 2023; Kumari et al., 2022; Ruiz et al., 2022). Besides text prompts, several works (Zhang & Agrawala, 2023; Mou et al., 2023) also introduce additional structure conditions to pre-trained text-to-image diffusion models for controllable text-to-image generation. Our ControlVideo is implemented based on the controllable text-to-image models to inherit their ability of high-quality and consistent generation.

**Text-to-video synthesis.** Large text-to-video generative models usually extend text-to-image models by adding temporal consistency. Earlier works (Wu et al., 2022a; Hong et al., 2022; Wu et al., 2021; Villegas et al., 2022) adopt an autoregressive framework to synthesize videos according to given descriptions. Capitalizing on the success of diffusion models in image generation, recent works (Ho et al., 2022a;b; Singer et al., 2022) propose to leverage their potential to produce high-quality videos. Nevertheless, training such large-scale video generative models requires extensive video-text pairs and computational resources. To reduce the training burden, Gen-1 (Esser et al., 2023) and Follow-Your-Pose (Ma et al., 2023) provide coarse temporal information (*e.g.*, motion sequences) for video generation, yet are still costly for most researchers and users. By replacing self-attention with the sparser cross-frame mechanisms, Tune-A-Video (Wu et al., 2022b) and Text2Video-Zero (Khacha-tryan et al., 2023) keep considerable consistency in appearance with little finetuning. ControlVideo also adapts text-to-image diffusion models without any training, but generates videos with better temporal consistency and continuity.

## 6 DISCUSSION

In this paper, we present a training-free framework, namely ControlVideo, towards consistent and continuous controllable text-to-video generation. ControlVideo, inflated from ControlNet, introduces an interleaved-frame smoother to ensure video continuity. Particularly, interleaved-frame smoother alternately smooths out the latents of three-frame clips, and stabilizes each clip by updating the middle frame with the interpolation among other two frames in latent space. Moreover, we empirically demonstrate the superior performance of fully cross-frame interaction, while presenting hierarchical sampler for long-video generation in commodity GPUs. Quantitative and qualitative experiments on extensive motion-prompt pairs demonstrate that ControlVideo achieves state-of-the-arts in terms of frame consistency and video continuity.

**Broader impact.** Large-scale diffusion models have made tremendous progress in text-to-video synthesis, yet these models are costly and unavailable to the public. ControlVideo focuses on training-free controllable text-to-video generation, and takes an essential step in efficient video creation. Concretely, ControlVideo could synthesize high-quality videos with commodity hardware, hence, being accessible to most researchers and users. For example, artists may leverage our approach to create fascinating videos with less time. Moreover, ControlVideo provides insights into the tasks involved in videoss, *e.g.*, video rendering, video editing, and video-to-video translation. On the flip side, albeit we do not intend to use our model for harmful purposes, it might be misused and bring some potential negative impacts, such as producing deceptive, harmful, or explicit videos. Despite the above concerns, we believe that they could be well minimized with some steps. For example, an NSFW filter can be employed to filter out unhealthy and violent content. Also, we hope that the government could establish and improve relevant regulations to restrict the abuse of video creation.

## ACKNOWLEDGEMENT

This work was supported by National Key RD Program of China under Grant No. 2021ZD0112100, and the National Natural Science Foundation of China (NSFC) under Grant No. U19A2073.

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

## A. DATASET DETAILS

In Table 4, we select 25 representative videos from DAVIS dataset (Pont-Tuset et al., 2017) and manually annotate their source captions. After that, we ask ChatGPT to generate five edited prompts for each source caption, following the instruction like: Please generate five new sentences that similar to "A man dances on the road", while being more diverse and highly detailed. Finally, we obtain 125 video-prompt pairs in total, and use them to evaluate both canny and depth conditioned generation.

## B. USER STUDY DETAILS

We conduct a user study to compare ControlVideo against two other methods on 125 samples, and ask five raters to answer questions in each sample. In Fig. 7, there are three questions involving in (i) video quality, (ii) temporal consistency, and (iii) text alignment. The raters are given unlimited time to make the selection. After collecting their answers, we take a majority vote as the final result for each sample, and present statistics in Table 2.

## C. MORE ABLATION STUDIES

During inference, we adopt DDIM sampling with $T = 50$ timesteps, which iteratively denoises a Gaussian noise from $T$ to $0$.

**Which timesteps does interleaved-frame smoother perform at?** In Fig. 8, we explore three timestep choices at different noise levels, including $\{48, 49\}$ at large noise level, $\{30, 31\}$ at middle noise level, and $\{0, 1\}$ at little noise level. When using the smoother at timesteps $\{48, 49\}$, the processed video is still unstable, since structure sequences bring additional flickers at the following timesteps. At timesteps $\{0, 1\}$ nearby image distribution, applying the interleaved-frame smoother leads to visible distortion in some frames. In contrast, performing smoothing operation at middle timesteps $\{30, 31\}$ promisingly deflickers the video, while preserving the quality and individuality of interpolated frames.

**How many timesteps are used in interleaved-frame smoother?** Fig. 9 shows the smoothed videos using interleaved-frame smoother at different numbers of timesteps. Applying the smoother at two consecutive timesteps (*i.e.*, 2 steps) could smooth the entire video with little video quality degradation. As the number of smoothing steps increases, the processed video is much smoother, but some frames become slightly blurred. Thus, for higher quality and efficiency, we set the number of smoothing timesteps as 2 by default.

**Non-deterministic DDPM-style sampler.** ControlVideo can also employ a non-deterministic DDPM-style sampler during inference. Following Eq.12 in DDIM (Song et al., 2020b), one can predict $z_{t-1}$ from $z_t$ via (*i.e.*, line 10 of Alg. 1 in paper):

$$z_{t-1} \leftarrow \sqrt{\alpha_{t-1}} \tilde{z}_{t\rightarrow 0} + \sqrt{1 - \alpha_{t-1}} \cdot \epsilon_\theta(z_t, t, c, \tau) + \sigma_t \epsilon_t, \tag{7}$$

where $\epsilon_t$ and $\sigma_t$ controls the level of random noise. DDPM results presents the generated videos of ControlVideo at different noise levels. Notably, as the noise level increases, ControlVideo generates more photo-realistic videos with dynamic details, *e.g.*, `ripples in the water`.

## D. MORE VISUALIZATIONS AND COMPARISONS

Fig. 10, Fig. 11, and Fig. 12 show more video visualizations conditioned on canny edges, depth maps, and human poses. Fig. 14, Fig. 15, and Fig. 16 present qualitative comparisons conditioned on canny edges, depth maps, and human poses. Fig. 13 provides an additional long video. More comparisons with video editing methods (Qi et al., 2023; Wang et al., 2023) are shown in this link.

Firstly, Vid2Vid-Zero and FateZero are designed for video editing by a hybrid of fully and sparse-casual cross-frame attention, and does not investigate different attention mechanisms in depth. In contrast, our ControlVideo focuses on continuous controllable text-to-video generation, and first empirically investigate the superiority of fully cross-frame attention. Secondly, Fig. 18 shows their qualitative comparisons on video editing. As one can see, the edited videos of ControlVideo not only have more consistent structure with source videos, but also aligns better with text prompts.

Table 4: **Names and captions of selected videos from DAVIS dataset.**

| Video Name | Source Caption |
|---|---|
| blackswan | a black swan moving on the lake |
| boat | a boat moves in the river |
| breakdance-flare | a man dances on the road |
| bus | a bus moves on the street |
| camel | a camel walks on the desert |
| car-roundabout | a jeep turns on a road |
| car-shadow | a car moves to a building |
| car-turn | a jeep on a forest road |
| cows | a cow walks on the grass |
| dog | a dog walks on the ground |
| elephant | an elephant walks on the ground |
| flamingo | a flamingo wanders in the water |
| gold-fish | golden fishers swim in the water |
| hike | a man hikes on a mountain |
| hockey | a player is playing hockey on the ground |
| kite-surf | a man is surfing on the sea |
| lab-coat | three women stands on the lawn |
| longboard | a man is playing skateboard on the alley |
| mallard-water | a mallard swims on the water |
| mbike-trick | a man riding motorbike |
| rhino | a rhino walks on the rocks |
| surf | a sailing boat moves on the sea |
| swing | a girl is playing on the swings |
| tennis | a man is playing tennis |
| walking | a selfie of walking man |

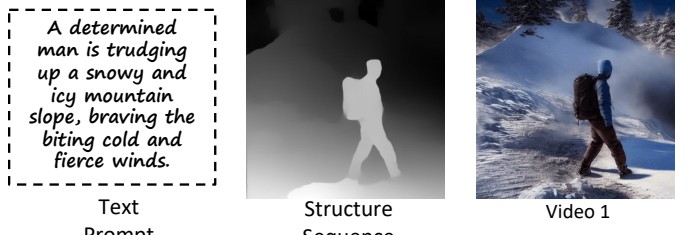

| Text Prompt | Structure Sequence | Video 1 | Video 2 |

*Between Method 1 & 2 :*
1. *Which video has higher quality ?*
2. *Which video has better temporal consistency across all frames?*
3. *Which video aligns better with text prompt?*

Figure 7: **The instruction of user study.** A user study sample consists of a text prompt, structure sequence, and synthesized videos from two different methods (in random order). The raters are asked to answer the above three questions for each sample.

## E. LIMITATIONS.

While our ControlVideo enables consistent and high-quality video generation, it still struggles with producing videos beyond input motion sequences. For example, in Fig. 17, given sequential poses of `Michael Jackson's moonwalk`, it is difficult to generate a vivid video according to text prompts like `Iron man runs on the street`. In this link, when input text prompts (*e.g.*, `rabbit`) seriously conflict with input motion (*e.g.*, ), the synthesized videos usually tend to align with input motion, ignoring the implicit structure in text prompts. To increase the ratio of text prompts over structure, we decrease the scale of ControlNet $\lambda$ to 0.3 ($\lambda = 1$ by default). Therefore, it can be seen $\lambda = 0.3$ that achieves a better trade-off between two input conditions than $\lambda = 1$. In the future, we will explore how to adaptively modify input motions according to text prompts, so that users can create more vivid videos.

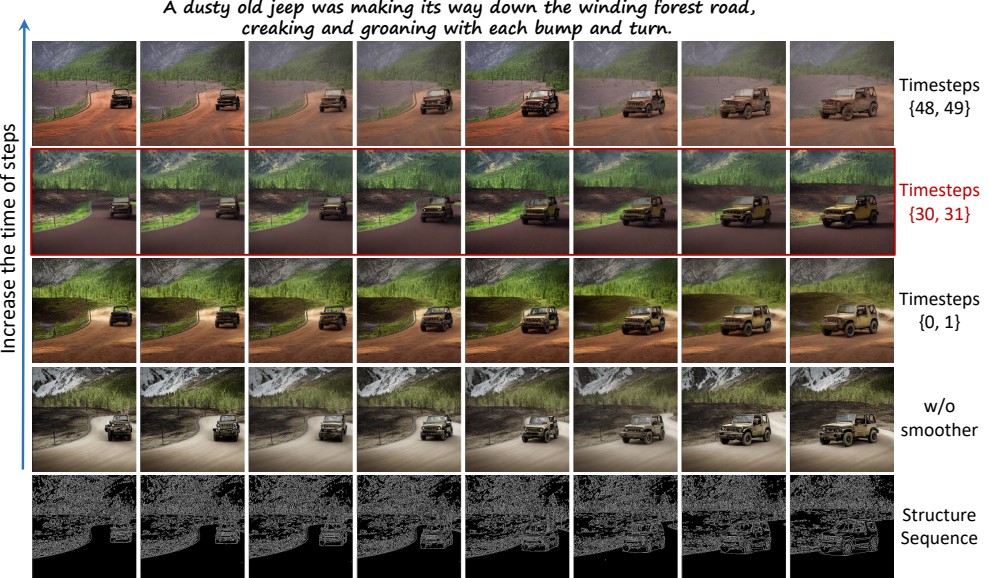

Figure 8: **Ablation on timestep choices in interleaved-frame smoother.** We apply interleaved-frame smoother at different timesteps, including $\{48, 49\}$ at large noise level, $\{30, 31\}$ at middle noise level, and $\{0, 1\}$ at little noise level. Among them, using the smoother at timesteps $\{30, 31\}$ promisingly mitigates the flicker effect while ensuring high quality. **Results best seen at 500% zoom.**

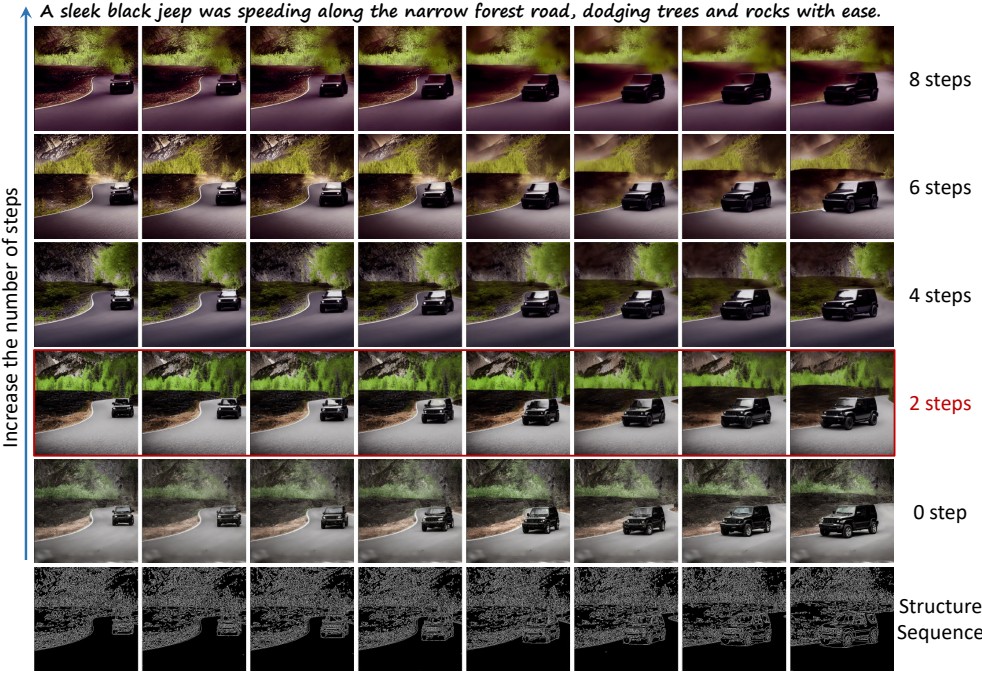

Figure 9: **Ablation on the number of timesteps used in interleaved-frame smoother.** Applying the smoother at two consecutive timesteps (*i.e.*, 2 steps) effectively reduces the flickers in structure. As we increase the number of smoothing steps, the processed video becomes smoother, but some frames are slightly blurred. Therefore, we set the number of smoothing steps as two by default. **Results best seen at 500% zoom.**

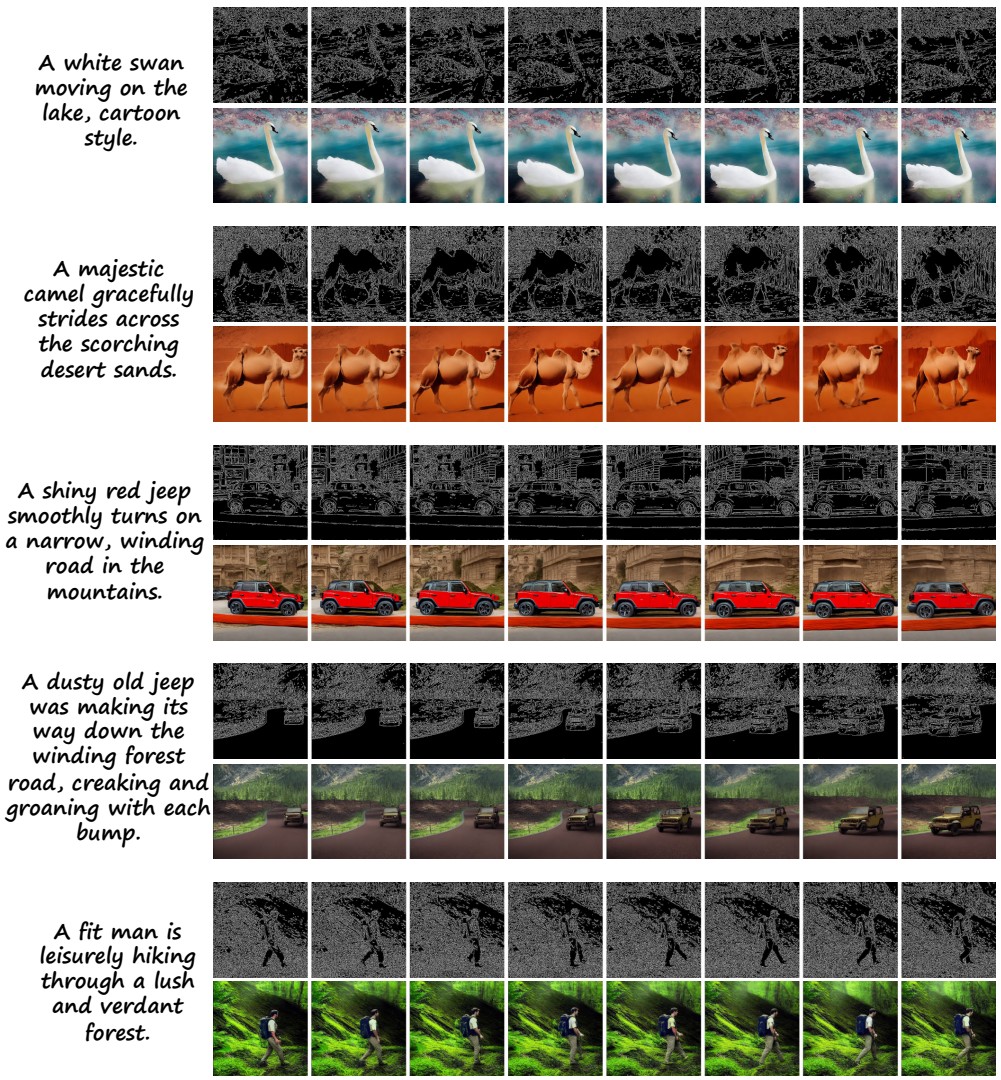

Figure 10: **More video visualizations conditioned on canny edges. Results best seen at 500% zoom.**

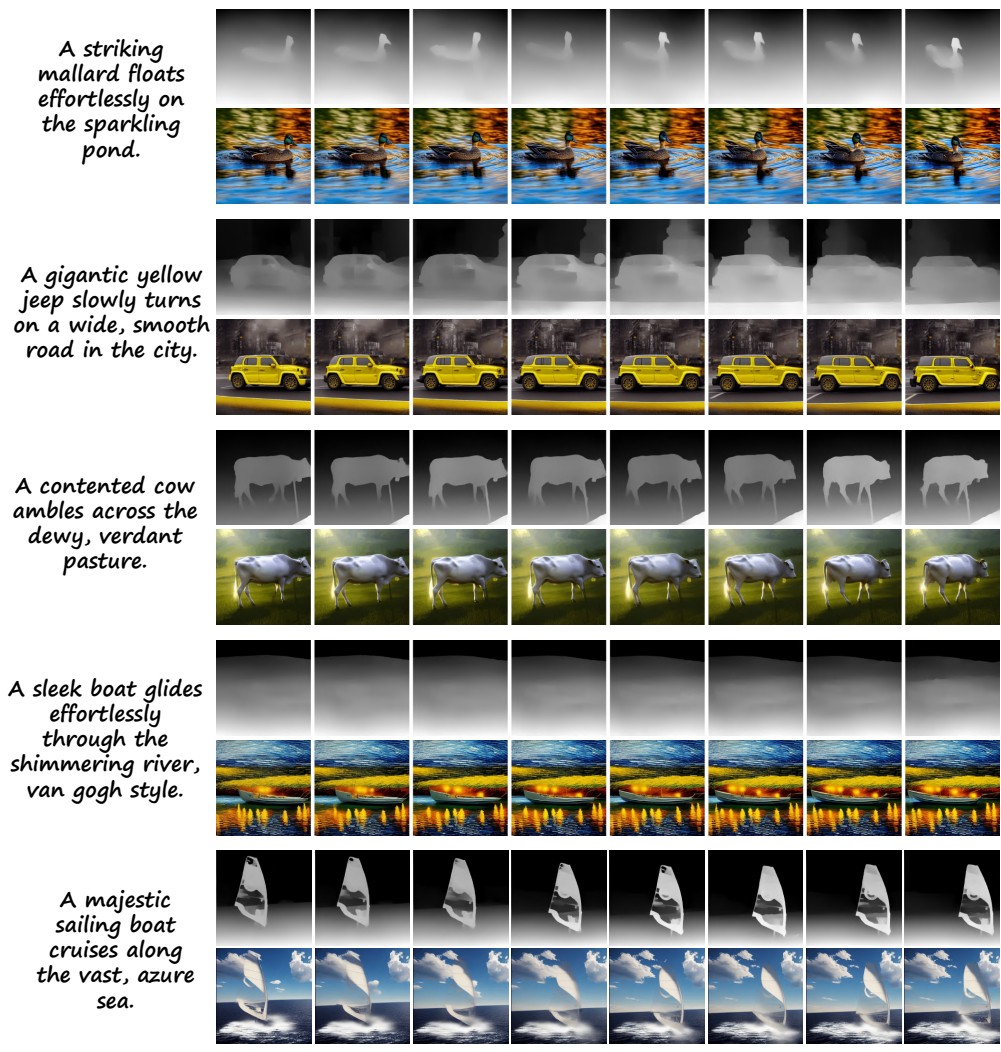

Figure 11: **More video visualizations conditioned on depth maps. Results best seen at 500%** **zoom.**

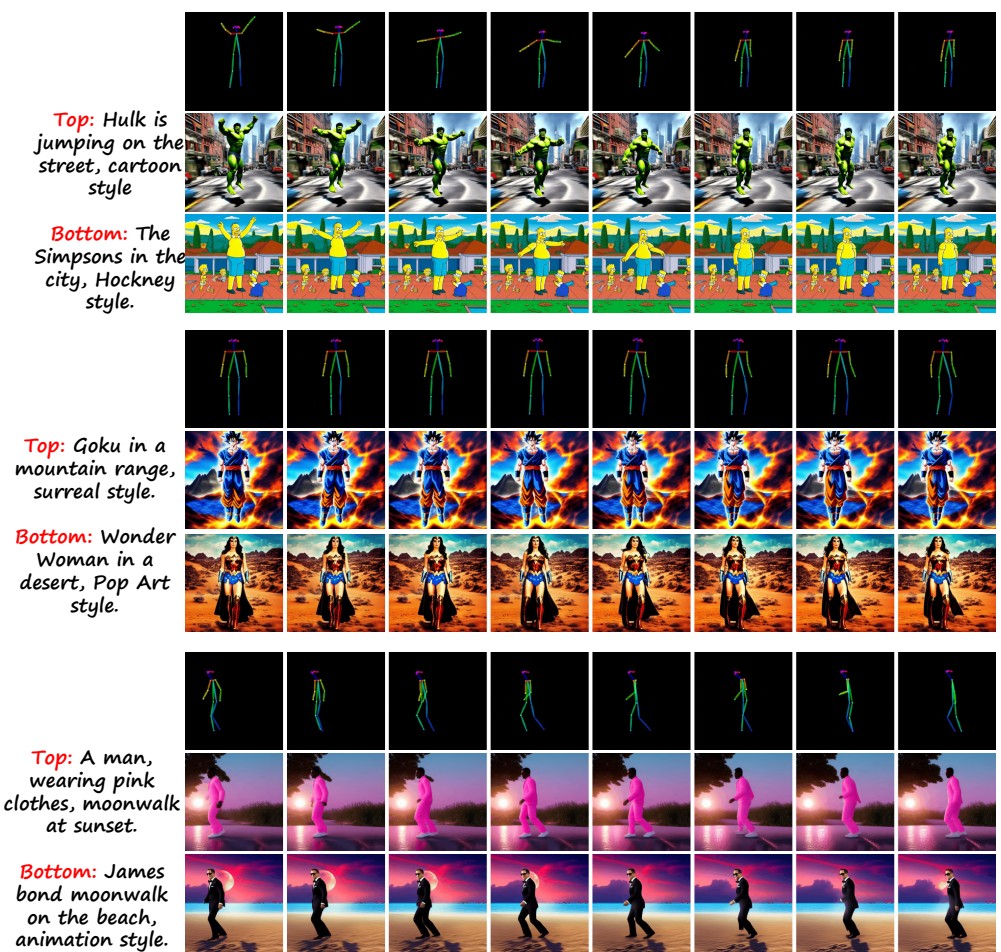

Figure 12: **More video visualizations conditioned on human poses. Results best seen at 500% zoom.**

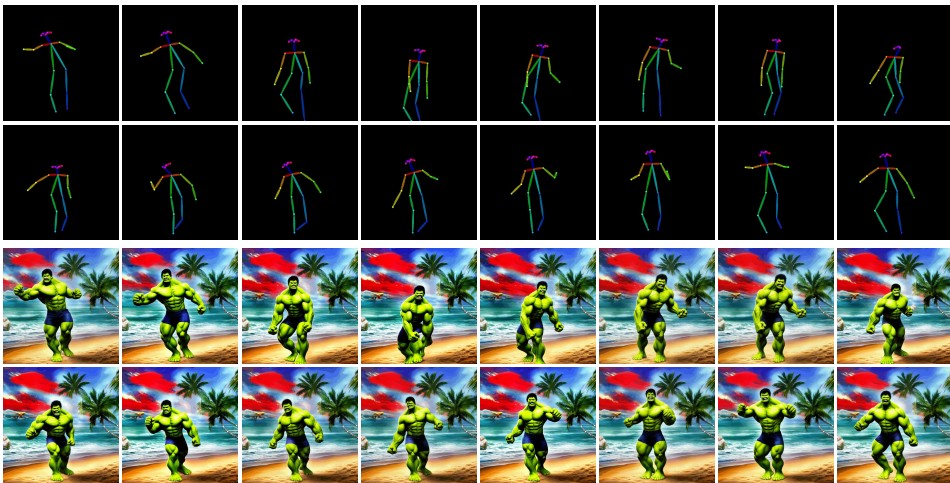

Hulk is dancing on the beach, cartoon style.

Figure 13: **Additional long video visualization. Results best seen at 500% zoom.**

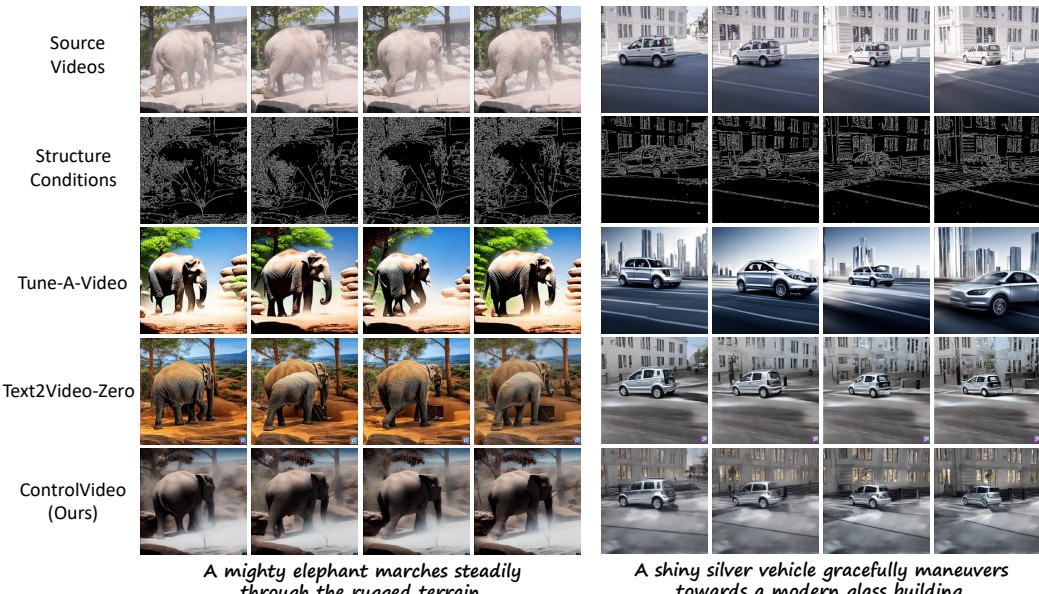

Figure 14: **More qualitative comparisons conditioned on canny edges. Results best seen at 500% zoom.**

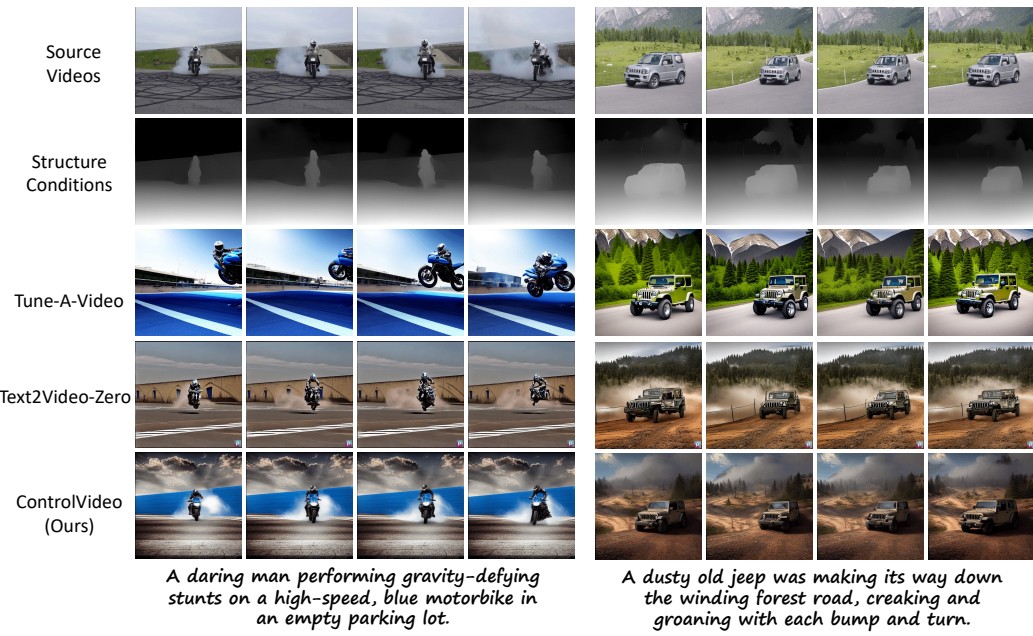

Figure 15: **More qualitative comparisons conditioned on depth maps. Results best seen at 500% zoom.**

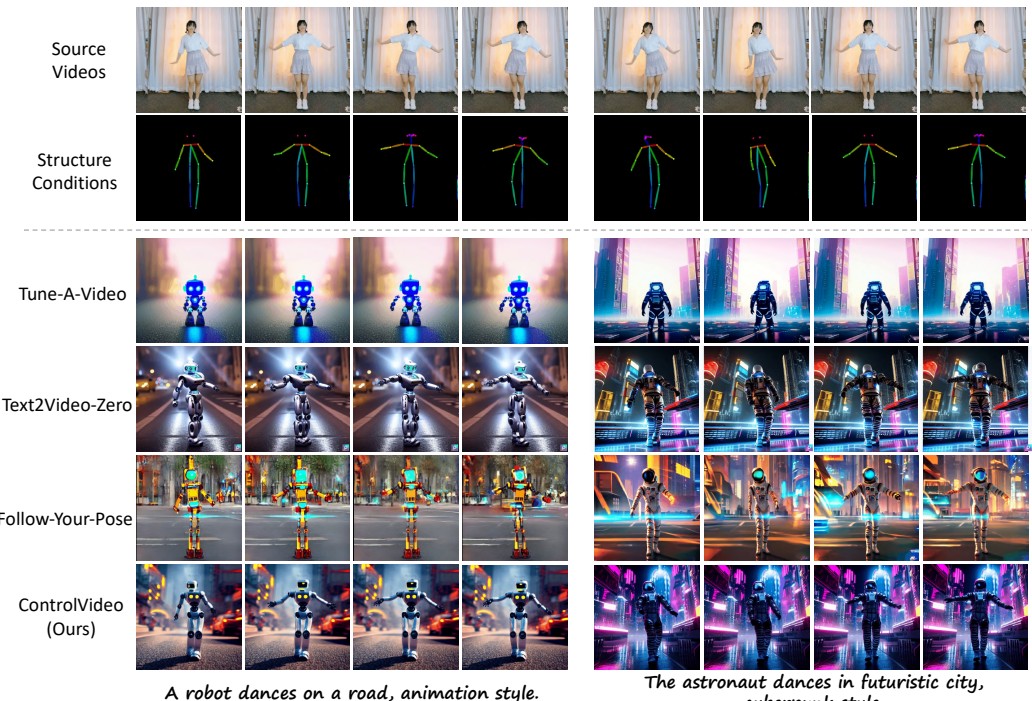

Figure 16: **More qualitative comparisons conditioned on human poses. Results best seen at 500% zoom.**

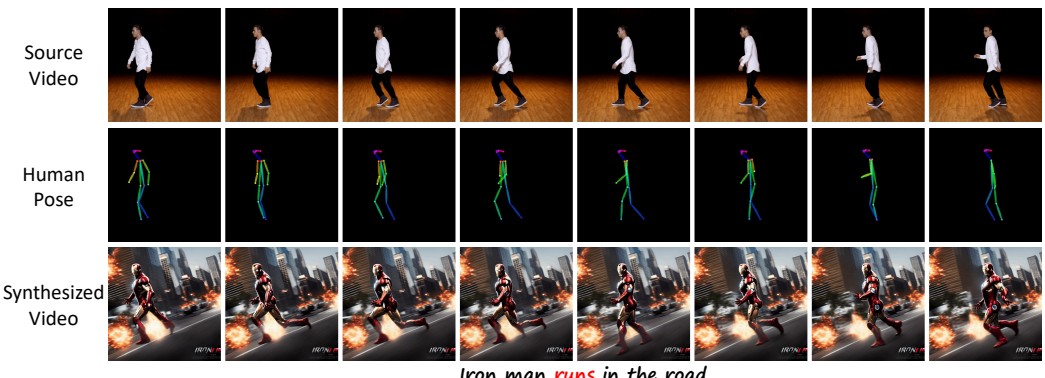

Figure 17: **Limitation visualizations.** ControlVideo struggles with producing videos beyond input motion sequences. The motion of text prompt `Iron man runs on the street` does not align with the given sequential poses of `Michael Jackson's moonwalk`, which degrades the video quality and consistency. *See videos at limitations.*

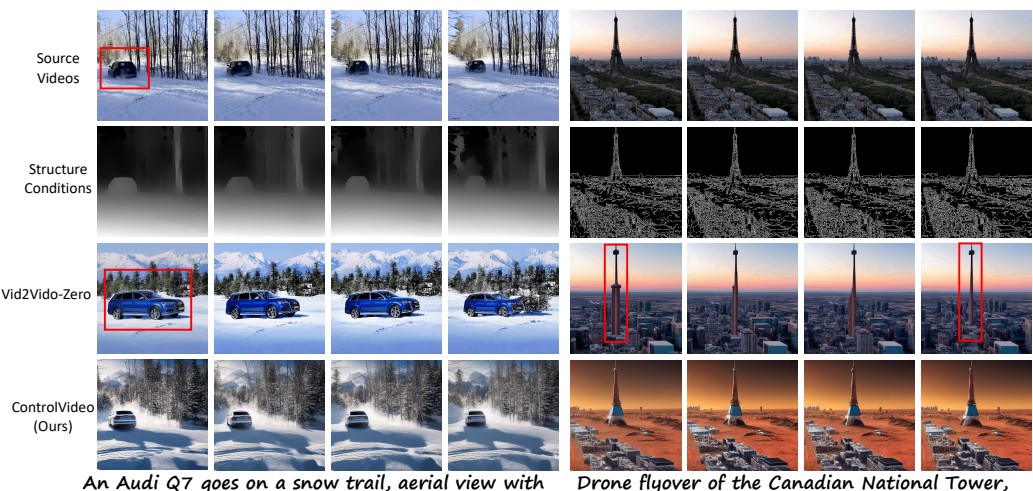

Figure 18: **Qualitative comparisons with Vid2Vid-Zero.** Inconsistent objects and prompts are colored in red.

