# OpenReview forum: "ControlVideo: Training-free Controllable Text-to-video Generation"
_ICLR.cc/2024/Conference — ICLR 2024 poster_

### Official Review · Reviewer_EvXG · 2023-10-29

**Soundness:** 3 good
**Presentation:** 3 good
**Contribution:** 3 good
**Rating:** 6
**Confidence:** 4

**Summary:**

This paper proposes a trainig-free framework to produce videos based on provided text prompts and motion sequences. An interleaved-frames smoother and a fully cross-frame interaction mechanism with a hierarchical sampler are proposed to enhance the quality of synthesized videos. The authors demonstrate that they achieve sota performance by entensive experiments.

**Strengths:**

1. The paper is well-written and easy-to-follow.
2. The proposed method is highly efficient and does not need training at all. Nevertheless, the quality of synthesized videos are not bad.

**Weaknesses:**

1. The proposed components (inter-frame interpolation, and cross-frame attention) are more-or-less explored in recent works. As such, I'm uncertain if this research will provide substantial insights to the community.

2. The proposed metrics (frame consistency, and prompt consistency) leverage CLIP model. Given that the CLIP model primarily operates in a deeply semantic and abstract domain, it often misses finer image details. Consequently, I'm inclined to think that the suggested metric might not adequately assess temporal consistency (for example, critizing jittering and discontinuity). Thus, the assertion that the proposed methods attain improved temporal consistency seems to lack robust quantitative backing.

3. This training-free framework cannot capture fine-grained motion pattern in video. Therefore, I believe it may not be the optimal approach for producing high-quality video content. Instead, I think finetuning the model on large-corpus video data might help improve the quality.

**Questions:**

Just as stated in the weakenss section, I am skeptical about the potential of training-free framework in video generation area. I'd be keen to hear the authors discuss potential future research directions in this direction.

---

> ### Author Response · Authors · 2023-11-20
> **Response to Reviewer EvXG**
>
> Dear reviewer, thanks for your insightful comments. We address the comments below and will include them in revision.
>
> **[W1]: The proposed components are more-or-less explored in recent works.**
>
> Our key insights lie in enhancing temporal consistency and continuity of videos, including three folds:
>
> **(i)** First of all, we introduce interleaved-frame smoother to alternately stabilize the video latents during sampling, and show that adding random noise produces more realistic videos ([[DDPM results](https://controlvideov1.github.io/#Non-deterministric%20DDPM%20sampler)], refer to [Q3 of Reviewer XMqd] for details)
>
> **(ii)** The hierarchical sampler achieves efficient long-video generation in commodity GPUs (See [[long-video results](https://controlvideov1.github.io/#Long%20video%20generation)]).
>
> **(iii)** We also empirically demonstrate the superior consistency and quality of fully cross-frame interaction (See [[cross-frame attention ablations](https://controlvideov1.github.io/#Effect%20of%20fully%20cross-frame%20interaction%20and%20interleaved-frame%20smoother)]).
>
> **[W2]: More robust metric for temporal consistency.**
>
> Following [1], we employ the optical-flow based warping error to robustly evaluate temporal continuity, where the optical flows comes from source videos.
> The following table reports the warping error comparisons of different methods.
> From the table, ControlVideo achieves a significantly lower warping error than other baselines, demonstrating its superiority in enhancing continuity.
> In addition, one can see that Canny-based models achieve lower warping error than depth-based ones, as Canny edges have more similar structure with optical flow.
>
> | Model                   | Condition       | Warping error ($\times 10^{-2}$) |
> | ----------------------- | --------------- | ---------------------- |
> | Tune-A-video            | DDIM inversion  | 18.16                  |
> | Text2Video-Zero         | Canny edges     | 8.76                   |
> | **ControlVideo (Ours)** | **Canny edges** | **2.75**               |
> | Text2Video-Zero         | Depth maps      | 10.36                  |
> | **ControlVideo (Ours)** | **Depth maps**  | **5.81**               |
>
>
> **[W3 & Q1]: The potential of training-free framework in video generation.**
>
> Albeit training on large-corpus videos can help improve the video consistency, there are three factors limiting further research:
> **(i)** Extremely high requirements for massive computational resources, which usually are unavailable for researchers from universities.
> **(ii)** Absence of sufficient video-caption datasets. Despite requiring modeling more complex video distributions, existing public video datasets only have limited number of videos compared to image datasets, e.g., WebVid-10M vs. LAION-5B.
> **(iii)** Lower frame quality than image generative models. Existing video datasets usually contain low-quality factors (e.g., blur and watermarks), and training on them may lead to visibly lower frame quality than the image counterparts.
>
> Due to limited computation resources, we investigate the effectiveness of proposed modules under the "training-free" setting, and demonstrate superior performance than some finetuning based methods, i.e., Tune-A-Video and Follow-Your-Pose.
> Since "training-free" ControlVideo has shown promisingly temporal consistency and continuity, it is natural to take it as a well-initialized model for the following finetuning.
> Compared to training from the scratch, it has two potential advantages:
> **(i)** Reduce the requirements for massive computation resources and data. The well-initialized model only needs to learn fine-grained motion pattern from videos.
> **(ii)** Alleviate low-quality effects from training videos. Requiring less finetuning in video dataset, ControlVideo may better preserve frame quality from the image counterparts.

---

> > ### Author Response · Authors · 2023-11-22
> > **Looking forward to further discussions**
> >
> > Dear reviewer, thank you again for your insightful comments on our paper, and we genuinely hope that our response could address your concerns. As the discussion is about to end, we are sincerely looking forward to your feedback. Please feel free to contact us if you have any further inquiries.

---

> > > ### Comment · Reviewer_EvXG · 2023-11-22
> > >
> > > Thanks the authors for clarification and providing results over additional metric. My concerns are addressed. Therefore, I would like to raise my score.

---

> > > > ### Author Response · Authors · 2023-11-22
> > > > **Thanks for your positive feedback**
> > > >
> > > > Thanks for your valuable suggestions and positive feedback on our work! They are very helpful for us to improve the paper, and we will include them in the revision.

---

### Official Review · Reviewer_aLwG · 2023-10-30

**Soundness:** 3 good
**Presentation:** 2 fair
**Contribution:** 2 fair
**Rating:** 5
**Confidence:** 5

**Summary:**

This work focuses on training-free controllable text-to-video generation tasks. It introduces an interleaved-frame smoother method to generate smoother frames. Additionally, it modifies cross-frame interaction to better utilize Stablediffusion's weights, enhancing frame continuity.

**Strengths:**

- The writing is clear and easy to follow.
- It is a training-free method, not relying on large-scale training, and has low computational resource requirements.
- The ablation experiments are well-designed and easy to understand.

**Weaknesses:**

- Overall, the innovation is average; applying ControlNet to video editing or generation is straightforward and easily thought of.
- The experiments are not comprehensive; there are too few baseline comparisons, and the experimental validation is limited to just over 20 examples, making the results less convincing.
- Limited by the absence of structure condition, this method can mainly edit videos with similar motion. Its effectiveness diminishes for videos with different motions or poses.

**Questions:**

As shown above, despite the method's average innovation and some shortcomings, I believe the exploration in this direction is worthwhile.
- I hope the authors can complete more experiments and cases, preferably providing an analysis of failure cases.
- Relying solely on the demo examples provided in the paper makes it challenging to be fully convinced.
- If the authors can address my concerns, I will consider giving a higher score.

**Details Of Ethics Concerns:**

no need

---

> ### Author Response · Authors · 2023-11-20
> **Response to Reviewer aLwG**
>
> Dear reviewer, thanks for your insightful comments. We hope our below replies could address the comments, and will include them in revision.
>
> **[W1]: Average innovation.**
>
> Albeit applying ControlNet to video generation is straightforward, our key insights lie in improving the temporal consistency and continuity, including three folds:
>
> **(i)** First of all, we introduce interleaved-frame smoother to alternately stabilize the video latents during sampling, and show that adding random noise produces more realistic videos ([[DDPM results](https://controlvideov1.github.io/#Non-deterministric%20DDPM%20sampler)], refer to [Q3 of Reviewer XMqd] for details)
>
> **(ii)** The hierarchical sampler achieves efficient long-video generation in commodity GPUs (See [[long-video results](https://controlvideov1.github.io/#Long%20video%20generation)]).
>
> **(iii)** We also empirically demonstrate the superior consistency and quality of fully cross-frame interaction (See [[cross-frame attention ablations](https://controlvideov1.github.io/#Effect%20of%20fully%20cross-frame%20interaction%20and%20interleaved-frame%20smoother)]).
>
> **[W2 & Q2]: Comparisons with more baselines.**
>
> Apart from original 30 examples, we have added 38 new examples to [[qualitative results](https://controlvideov1.github.io/#Qualitative%20comparisons)], including another two baselines of video editing (FateZero[1] and Vid2Vid-Zero[2]).
> From [[qualitative results](https://controlvideov1.github.io/#Qualitative%20comparisons)], ControlVideo generates videos with better consistency and continuity than other approaches.
> In contrast, FateZero may fail to align with text prompts, e.g., Iron man in [[human-pose results](https://controlvideov1.github.io/#Human%20pose)], while Vid2Vid-zero produces videos with more visible flickers.
>
> **[W3 & Q1]: Failure cases analysis.**
>
> We provide two additional failure cases in [[failure-case results](https://controlvideov1.github.io/#Trade-off%20between%20text%20prompt%20and%20motion)].
> As one can see, when input text prompts (e.g., rabbit) seriously conflict with input motion (e.g., cow), the synthesized videos usually tend to align with input motion, ignoring the implicit structure in text prompts.
> To increase the ratio of text prompts over structure, we decrease the scale of ControlNet $\lambda$ to 0.3 ($\lambda=1$ by default).
> In [[failure-case results](https://controlvideov1.github.io/#Trade-off%20between%20text%20prompt%20and%20motion)], it can be seen that $\lambda=0.3$ achieves a better trade-off between two input conditions than $\lambda=1$.
> In the future, we will explore how to adaptively modify input motions according to text prompts, so that users can create more vivid videos.
>
> [1] Chenyang Qi, et al. FateZero: Fusing Attentions for Zero-shot Text-based Video Editing. In ICCV 2023.
>
> [2] Wen Wang, et al. Zero-Shot Video Editing Using Off-The-Shelf Image Diffusion Models. Arxiv 2023.

---

> > ### Author Response · Authors · 2023-11-22
> > **Looking forward to further discussions**
> >
> > Dear reviewer, thank you again for your insightful comments on our paper, and we genuinely hope that our response could address your concerns. As the discussion is about to end, we are sincerely looking forward to your feedback. Please feel free to contact us if you have any further inquiries.

---

> > > ### Author Response · Authors · 2023-11-23
> > > **Looking forward to further discussions**
> > >
> > > Dear reviewer, thank you again for your invaluable comments. We have conducted adequate experiments during rebuttal and sincerely hope that they could address your concerns. As the discussion is about to end today, we are sincerely looking forward to your feedback. Please feel free to contact us if you have any further inquiries.

---

### Official Review · Reviewer_HJ3a · 2023-11-01

**Soundness:** 3 good
**Presentation:** 3 good
**Contribution:** 2 fair
**Rating:** 6
**Confidence:** 4

**Summary:**

This paper proposes ControlVideo, a training-free framework that can produce high-quality videos based on the provided text prompts and motion sequences (e.g., different modalities). ControlVideo adapts a pre-trained text-to-image model (i.e., ControlNet) for controllable text-to-video generation. The paper introduces an interleaved-frame smoother that alternately smooths out the latents of successive three-frame clips by updating the middle frame with the interpolation among the other two frames in the latent space, aiming to stabilize the temporal continuity of the generated videos. Besides, a fully cross-frame interaction mechanism is exploited to further enhance the frame
consistency, and a hierarchical sampler is employed to produce long videos more efficiently. Experimental results demonstrate that the proposed ControlVideo outperforms the state-of-the-art baselines both quantitatively and qualitatively.

**Strengths:**

- The paper is clearly written, well organized, and easy to follow. The symbols, terms, and concepts are adequately defined and explained. The language usage is good.

- The proposed method is simple and easy to understand. Sufficient details are provided for the readers.

- The experiments are generally well-executed. The empirical results show the effectiveness of the proposed method, showing certain advantages over state-of-the-art baselines.

**Weaknesses:**

- The qualitative results showcase certain advantages of the proposed method over state-of-the-art baselines in controllable text-to-video generation. However, by checking the provided video results, the temporal consistency can still be improved. Also, in some cases, the background looks unchanged. Some visual details can still be improved. Providing more discussions on these could strengthen this paper further.

- The fully cross-frame interaction mechanism considers all frames as the reference, which thus increases the computational burden. What is the intuition to consider all the frames as a large image? Why not select some key frames to reduce redundant information? It is interesting to provide more discussions and analysis on this.

- The paper mentioned that the proposed interleaved-frame smoother is performed on predicted RGB frames at timesteps {30, 31} by default. It can be more interesting if more studies and analyses on different steps to apply such a mechanism are provided.

- It seems the interleaved-frame smoother still brings more computational cost and affects the model efficiency due to the additional conversion and interpolation steps.

**Questions:**

- Why does the hierarchal sampler improve model efficiency? It seems all the frames still need to be generated, although it is a top-down generation from key frames.

- It is suggested to remove some content about the background and preliminary since such information is well-known.

- The reviewer is interested if the proposed ControlVideo can be extended to generate more challenging new information, such as a novel view/unseen part of an object.

-  Will the authors release all the code, models, and data to ensure the reproducibility of this work?

---

> ### Author Response · Authors · 2023-11-20
> **Response to Reviewer HJ3a (Part 1/2)**
>
> Dear reviewer, thanks for your insightful comments.We provide our replies to the comments below, and hope they could address your concerns.
>
> **[W1]: Temporal consistency to be improved and unchanged background.**
>
> **(i) Temporal inconsistency.**
> For some videos containing flickers, there are two potential reasons:
> (a) The input motion contains visible flickers and may result in the flickers in synthesized videos.
> (b) Sometimes, the scene correspondence in motions cannot be well maintained through cross-frame attention only, leading to incoherence in appearance.
> For (a), we can leverage video-based annotator[1] or smoothing filters[2] to obtain more stable motion sequence.
> For (b), one may first estimate the correspondence of motions with[3,4], and then recalibrate the attention weights based on the correspondence.
>
>
> **(ii) Unchanged background.**
> In such cases, the background part of input motion is unchanged, and its synthesized background will also be static via DDIM sampling.
> As shown in [[DDPM results](https://controlvideov1.github.io/#Non-deterministric%20DDPM%20sampler)], when ControlVideo employs non-deterministic DDPM sampling, it produces more vivid videos with lively background.
> Kindly refer to [Q3 of Reviewer XMqd] for more details.
>
> **[W2]: Consider all frames as a large image and select some key frames**
>
> **(i)**
> For original self attention in Stable Diffusion (SD), both its queries and keys come from the same image.
> Similarly, fully cross-frame attention computes queries and keys with the same frames, and may better inherit high-quality and consistent generation than other cross-frame mechanisms.
> **(ii)** We conduct ablation experiments on different number $k$ of key frames, and the results are given in the following table and [[keyframes results](https://controlvideov1.github.io/#Effect%20of%20fully%20cross-frame%20interaction%20and%20interleaved-frame%20smoother)].
> With the increasing of $k$, ControlVideo obtains better performance in temporal consistency, i.e., higher frame consistency and lower warping error.
> Also, larger $k$ visibly reduces frame incoherence in appearance, e.g., the orientation of elephant.
>
> | Number of keyframes                     | k=1 (first-only) | k=2 (sparse-casual) | k=4   | k=8   | k=15 (fully) | k=15  w/ smoother |
> | --------------------------------------- | ---------------- | ------------------- | ----- | ----- | ------------ | ----------------- |
> | **Frame consistency $\uparrow$**        | 94.92            | 95.06               | 95.18 | 95.30 | `95.36`      | **96.83**         |
> | **Warping error ($\times 10^{-2}$) $\downarrow$** | 8.91             | 7.05                | 6.62  | 6.27  | `5.93`       | **2.75**          |
> | **Time cost (min)$\downarrow$**         | 1.2              | 1.5                 | 1.9   | 2.5   | 3.0          | 3.5               |
>
> **[W3]: Apply smoother at different timesteps.**
>
> The following table and [[smoother-timesteps results](https://controlvideov1.github.io/#Which%20timesteps%20does%20interleaved-frame%20smoother%20perform)] show ablation experiments in different timesteps (refer to Appendix C for more details).
> As one can observe, applying the smoother at timesteps {30,31} effectively deflickers the video while ensuring its frame quality.
> In contrast, using the smoother at timesteps {48,49} or timesteps {0,1} lead to slight flickering or distortion, degrading the effectiveness of smoother.
>
> | Timestep choices                        | {}    | {48,49} | {30,31}   | {20,21} | {10,11} | {0,1} |
> | --------------------------------------- | ----- | ------- | --------- | ------- | ------- | ----- |
> | **Frame consistency $\uparrow$**        | 95.36 | 95.97   | **96.83** | 96.82   | 96.81   | 96.78 |
> | **Warping error ($\times 10^{-2}$) $\downarrow$** | 5.93  | 4.87    | **2.75**  | 2.92    | 3.24    | 3.89  |
>
> **[W4]: More computational cost of smoother.**
>
> From above table and [[cross-frame attention and smoother](https://controlvideov1.github.io/#Effect%20of%20fully%20cross-frame%20interaction%20and%20interleaved-frame%20smoother)], the proposed smoother greatly improves temporal continuity of synthesized videos, and the extra 0.5 min can be seen as acceptable.
>
> **[Q1]: Why does the hierarchical sampler improve model efficiency?**
> The hierarchical sampler is designed to improve efficiency of fully cross-frame attention, especially in producing long videos.
> Given $N$ frames, $K$ key frames and $T$ tokens for each frame, the complexity for fully attention is $O(T^2 \cdot N^2)$, while that of hierarchical sampler is $O(T^2 \cdot (K^2 + \frac{N^2}{K}))=O(T^2 \cdot (N + N^{\frac{3}{2}}))$ ($K=\sqrt{N}$ by default).
> When producing a video of 100 frames, the fully attention requires about 50 mins while the hierarchical sampler only takes about 10 mins.

---

> > ### Author Response · Authors · 2023-11-20
> > **Response to Reviewer HJ3a (Part 2/2)**
> >
> > **[Q2]: Remove some content about the background and preliminary.**
> > Thanks for your kind suggestion, we will simplify the background and preliminary parts to make the revised paper more cohesive.
> > Specifically, we will remove most content of LDM and only preserve the content related to DDIM, since it will be used in the interleaved-frame smoother.
> >
> >
> > **[Q3]: Extend to novel view/unseen part of an object.**
> > From our understanding, we interpreted this question as asking for producing videos with different views of an object.
> > Please let us know if we have misinterpreted it.
> > Considering that ControlVideo requires structure sequence as input, we leverage the depth map sequence as different views.
> > From [[novel-view results](https://controlvideov1.github.io/#Novel%20view%20generation)], one can see that ControlVideo is able to produce consistent views of an object.
> >
> > **[Q4]: Release code, models and data.**
> > Yes, we are sure to make all the code, models, and data public after accepted.
> >
> > [1] Xuan Luo, et al. Consistent Video Depth Estimation. In SIGGRAPH 2020.
> >
> > [2] Press W.H, et al. Savitzky-Golay smoothing filters. In Computers in Physics.
> >
> > [3] David Lowe, et al. Distinctive image features from scale-invariant keypoints. In IJCV 2004.
> >
> > [4] Luming Tang, et al. Emergent Correspondence from Image Diffusion. In NeurIPS 2023.

---

> > > ### Author Response · Authors · 2023-11-22
> > > **Looking forward to further discussions**
> > >
> > > Dear reviewer, thank you again for your insightful comments on our paper, and we genuinely hope that our response could address your concerns. As the discussion is about to end, we are sincerely looking forward to your feedback. Please feel free to contact us if you have any further inquiries.

---

> > > > ### Author Response · Authors · 2023-11-23
> > > > **Looking forward to further discussions**
> > > >
> > > > Dear reviewer, thank you again for your invaluable comments. We have conducted adequate experiments during rebuttal and sincerely hope that they could address your concerns. As the discussion is about to end today, we are sincerely looking forward to your feedback. Please feel free to contact us if you have any further inquiries.

---

### Official Review · Reviewer_XMqd · 2023-11-02

**Soundness:** 2 fair
**Presentation:** 2 fair
**Contribution:** 2 fair
**Rating:** 5
**Confidence:** 5

**Summary:**

This paper introduces "ControlVideo," a training-free framework that significantly improves text-driven video generation. It addresses issues like appearance inconsistency and flickers in long videos through innovative modules for frame interaction and smoothing. ControlVideo outperforms existing methods, efficiently generating high-quality videos within minutes.

**Strengths:**

•	The proposed method is straightforward, easily implementable, and reproducible, making it accessible for further research and application.

•	The paper introduces novel techniques for long video generation, and the "interleaved-frame smoother" effectively improves frame consistency.

•	The results demonstrate improvements over existing methods, substantiating the paper's claims.

**Weaknesses:**

•	While the full-attention mechanism and "interleaved-frame smoother" enhance frame consistency, they also significantly increase the computational time.

•	The background appears to flicker in relation to the foreground in some examples. For instance, in the "James Bond moonwalk on the beach, animation style" video on the provided website, the moon inconsistently appears and disappears.

•	The paper lacks quantitative comparisons with Text2Video-Zero in the context of pose conditions, which could be a significant oversight given the importance of pose in video generation.

**Questions:**

•	Could you provide additional results for long video generation to further validate the method's efficacy?

•	Is there a potential solution to the flickering background issue mentioned in the second weakness?

•	Would it be possible to employ a non-deterministic DDPM-style sampler as an alternative to DDIM?

---

> ### Author Response · Authors · 2023-11-20
> **Response to Reviewer XMqd**
>
> Dear reviewer, thanks for your insightful comments. We provide our replies to the comments below, and hope they could address your concerns.
>
>
> **[W1]: Increasing computational time.**
>
> The following table and [[cross-frame attention and smoother](https://controlvideov1.github.io/#Effect%20of%20fully%20cross-frame%20interaction%20and%20interleaved-frame%20smoother)] show ablation experiments of the fully cross-frame attention and interleaved-frame smoother.
> As one can see, the proposed two modules significantly improve both video consistency and continuity, with acceptable increase in computational time.
>
> | Cross-frame attention   | Frame consistency $\uparrow$ | Warping error ($\times 10^{-2}$) $\downarrow$ | Time cost (min)$\downarrow$ |
> | ----------------------- | ---------------------------- | ----------------------------------- | --------------------------- |
> | First-only              | 94.92                        | 8.91                                | 1.2                         |
> | Sparse-causal           | 95.06                        | 7.05                                | 1.5                         |
> | Fully                   | 95.36                        | 5.93                                | 3.0                         |
> | Fully + Smoother (Ours) | **96.83**                    | **2.75**                            | 3.5                         |
>
>
> **[W2 & Q2]: Flickering background issue.**
>
> For some cases with flickering background, there are two potential reasons:
> (a) The input motion contains visible flickers and may bring flickering controls to synthesized background.
> (b) Sometimes, the scene correspondence in motions cannot be well maintained through cross-frame attention only, producing videos with incoherent appearance.
> For (a), we can leverage video-based annotator[3] or smoothing filters [4] to obtain more stable motion sequence.
> For (b), we first estimate the correspondence of motions with [5,6], and then recalibrate the attention weights based on the correspondence.
>
> **[W3]: Quantitative comparisons with Text2Video-Zero in pose conditions.**
>
> In terms of pose condition, we compare our ControlVideo with Text2Video-Zero in the test set of Fashion-Text2Video dataset[1], which contains 380 video-prompt pairs in total.
> From the following table, ControlVideo is superior to Text2Video-Zero by frame consistency, while with marginal gains in prompt consistency.
>
> | Model               | Condition | Frame consistency | Prompt consistency |
> | ------------------- | --------- | ----------------- | ------------------ |
> | Text2Video-Zero     | Pose      | 91.85             | 28.86              |
> | ControlVideo (Ours) | Pose      | **93.24**         | **28.94**          |
>
>
> **[Q1]: Additional results for long video generation.**
>
> In addition to the original two long videos, we have added another three ones to [[long-video results](https://controlvideov1.github.io/#Long%20video%20generation)].
> Benefiting from our hierarchical sampler, it only takes $\sim$ 10 minutes to generate a video with 100 frames in one NVIDIA RTX 2080Ti.
>
> **[Q3]: Non-deterministic DDPM-style sampler.**
>
> Yes, our ControlVideo can also employ a non-deterministic DDPM-style sampler during inference.
> Following Eq.12 in DDIM[2], one can predict $z_{t-1}$ from $z_t$ via (i.e., line 10 of Alg.1 in paper):
>
> $z\_{t-1} = \sqrt{\alpha\_{t-1}} \tilde{z}\_{t\rightarrow 0} + \sqrt{1 - \alpha\_{t-1} - \sigma\_t^2} \cdot \epsilon\_\theta(z_t, t, c, \tau) + \sigma\_t \epsilon_t$.
>
> where $\epsilon\_t$ is random Gaussian noise and $\sigma\_t = \lambda \cdot \sqrt{(1 - \alpha\_{t-1})/(1 - \alpha\_{t})}\sqrt{1 - \alpha\_{t}/\alpha\_{t-1}}$ controls the level of random noise.
>
> [[DDPM results](https://controlvideov1.github.io/#Non-deterministric%20DDPM%20sampler)] presents the generated videos of ControlVideo at different noise levels.
> Notably, as the noise level increases, ControlVideo generates more photo-realistic videos with dynamic details, e.g., ripples in the water.
>
> [1] Yuming Jiang, et al. Text2Performer: Text-Driven Human Video Generation. In ICCV 2023.
>
> [2] Jiaming Song, et al. Denoising Diffusion Implicit Models. In ICLR 2021.
>
> [3] Xuan Luo, et al. Consistent Video Depth Estimation. In SIGGRAPH 2020.
>
> [4] Press W.H, et al. Savitzky-Golay smoothing filters. In Computers in Physics.
>
> [5] David Lowe, et al. Distinctive image features from scale-invariant keypoints. In IJCV 2004.
>
> [6] Luming Tang, et al. Emergent Correspondence from Image Diffusion. In NeurIPS 2023.

---

> > ### Author Response · Authors · 2023-11-22
> > **Looking forward to further discussions**
> >
> > Dear reviewer, thank you again for your insightful comments on our paper, and we genuinely hope that our response could address your concerns. As the discussion is about to end, we are sincerely looking forward to your feedback. Please feel free to contact us if you have any further inquiries.

---

> > > ### Author Response · Authors · 2023-11-23
> > > **Looking forward to further discussions**
> > >
> > > Dear reviewer, thank you again for your invaluable comments. We have conducted adequate experiments during rebuttal and
> > >  sincerely hope that they could address your concerns. As the discussion is about to end today, we are sincerely looking forward to your feedback. Please feel free to contact us if you have any further inquiries.

---

### Meta-Review · Area_Chair_Q71R · 2023-12-10

**Metareview:**

The paper received borderline scores. The reviewers suggested lack of novelty, insufficient comparisons, flickering background. During the discussion stage, the authors have provided the necessary comparisons and details, so that one of the reviewers raised their score slightly. While the AC agrees with the reviewers, the AC also thinks that the paper can be a valuable contribution. Training-free methods are important to explore as the computational requirements go up. Hence, the AC decided to accept the manuscript. Congrats!

**Justification For Why Not Higher Score:**

This is a borderline paper, it shouldn't get higher score.

**Justification For Why Not Lower Score:**

We can give lower score here. It'll be okay. But AC thought that the paper will be valuable for the community.

---

### Decision · Program_Chairs · 2024-01-16

Accept (poster)